# LoRa Channel Characterization for Flexible and High Reliability Adaptive Data Rate in Multiple Gateways Networks

Ulysse Coutaud *, Martin Heusse and Bernard Tourancheau 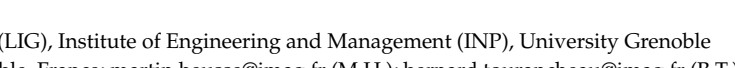

Grenoble Informatic Laboratory (LIG), Institute of Engineering and Management (INP), University Grenoble Alps (UGA), CNRS, 38058 Grenoble, France; martin.heusse@imag.fr (M.H.); bernard.tourancheau@imag.fr (B.T.)
* Correspondence: ulysse.coutaud@univ-grenoble-alpes.fr

**Abstract:** We characterize the LoRa channel in terms of multi-path fading, loss burstiness, and assess the benefits of Forward Error Correction as well as the influence of frame length. We make these observations by synthesizing extensive experimental measurements realized with The Things Network in a medium size city. We then propose to optimize the LoRaWAN Adaptive Data Rate algorithm based on this refined LoRa channel characterization and taking into account the LoRaWAN inherent macro-diversity from multi-gateway reception. Firstly, we propose $ADR_{opt}$, which adjusts Spreading Factor and frame repetition number to maintain the communication below a target Packet Error Rate ceiling with optimized Time-On-Air. Secondly, we propose $ADR_{IFECC}$, an extension of $ADR_{opt}$ in case an Inter-Frame Erasure Correction Code is available. The resulting protocol provides very high reliability even over low quality channels, with comparable Time on Air and similar downlink usage as the currently deployed mechanism. Simulations corroborate the analysis, both over a synthetic random wireless link and over replayed real-world packet transmission traces.

**Keywords:** LoRaWAN; macro-diversity; ADR; reliability; characterization; IoT; LPWAN; LoRa





## 1. Introduction

The growth of the Internet of Things (IoT) brings legacy wireless networks technologies to their limits. The scalability, energy consumption and cost of conventional cellular technologies makes them unsuitable for the massive deployments required in the contexts of the smart city, smart farm, smart factory, wide scale asset tracking, and so forth. To address these challenges, Low Power Wide Area Networks (LPWAN) promise to provide long range and large scale connectivity for the IoT, at low cost and low power consumption.

LoRaWAN® is one of the leading LPWAN technologies, it is a networking protocol specification developed by the open LoRa Alliance® on top of Semtech's proprietary modulation LoRa®. With a large physical and MAC parameters set, LoRaWAN provides high flexibility to optimize the network performances in terms of energy consumption, communication range, reliability, throughput or scalability. So, LoRaWAN claims the ability to provides connectivity to thousands of battery-powered autonomous End-Devices (EDs) within ten kilometers of a single gateway (GW) with throughput up to a kilobyte per second, for a decade. These performances attract the interest of both the academic community and the industry, and put LoRaWAN at the forefront among LPWAN technologies.

But versatility comes at the cost that LoRaWAN requires careful engineering to get the best of the technology, or the performance quickly falls below expectations. EDs deployed in the LoRaWAN network need to be configured according to the actual channel condition and network load in order to reach maximal performances and meet the applications requirements. Moreover, precise LoRaWAN channel characterization in a deployed network is a fundamental need to strive for maximum performances in real conditions.

In this work, we start with the analysis of experimental measurements over a public LoRaWAN network with multiple reception gateways. From the insight gained into the

channel characteristics, we derive methods to use the network with optimized performances in terms of reliability and Time-On-Air.

This paper is organized as follows: Section 2 presents LoRa, LoRaWAN, the metrics to measure performance and then the experimental traces database construction. Section 3 presents the experimental channel characterization. Section 4 presents the transmission parameters selection for network-wide optimization. Section 5 discusses the related work. Section 6 concludes the article.

## 2. Materials and Methods

### 2.1. LoRaWAN Protocol Stack

#### 2.1.1. LoRa Physical Layer

The LoRa modulation [1] is based on Compressed High Intensity Radar Pulse (CHIRP) Spread Spectrum (CSS). A LoRa frame is composed of a series of chirp symbols. Each symbol is a linearly increasing frequency ramp mapped cyclically over the radio channel bandwidth (*BW*).

The information is encoded by the chirp initial frequency offset. As illustrated in Figure 1, the spreading factor (*SF*) defines the symbol duration, as $T_{\text{symbol}} = \frac{2^{SF}}{BW}$. Each symbol conveys *SF* bits. In the current LoRa implementations, SF 6 to 12 are available. A higher *BW* increases frequency spreading but reduces time spreading that is, the symbol duration, resulting in an increased data rate. A higher *SF* increases the symbol duration, reduces the data rate and makes the modulation more robust.

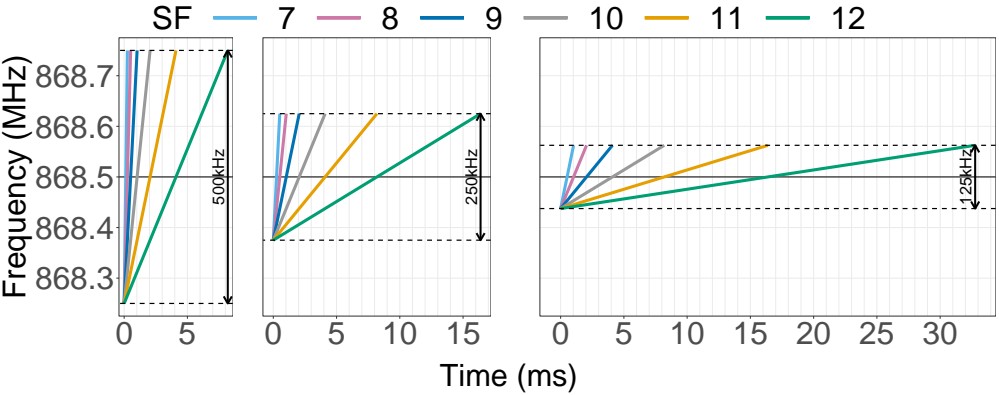

**Figure 1.** LoRa CHIRPs for SF $\in [7..12]$ and BW $\in \{125, 250, 500\}$ kHz on the 868.5 MHz channel.

The inherently robust CSS modulation scheme is complemented by an intra-packet error correcting code with coding rate (*CR*) between $\frac{4}{5}$ and $\frac{4}{8}$. The LoRa physical layer has a very high link budget of up to 153.5 dB (with Semtech SX1301 chip and 14 dBm transmission power ($P_{\text{Tx}}$)) and it is robust against noise, Doppler effect and frequency drift, which allows the use of less precise, thus cheaper, hardware. Changing the physical layer transmission parameters (*BW*, *CR*, *SF*) allows to trade robustness (i.e., link budget) for data rate, which is inversely proportional to the time on air for a given frame size.

#### 2.1.2. LoRaWAN

LoRaWAN [2] is an LPWAN protocol stack build on top of the LoRa [1] physical layer. The LoRaWAN uplink frame structure is given in Figure 2. The network topology is cellular-like with several gateways covering the area of interest, often with overlapping coverage zones. The LoRaWAN gateways (GW) relay End Devices (EDs) uplink messages to a central network server (NS). EDs are not associated to a particular GW—the GWs forward all received messages to the NS, and uplink traffic thus benefits from GW diversity. Most of the network complexity is pushed to the NS which handles messages de-duplication, downlink scheduling and routing of uplink data to the application servers. The channel access method is ALOHA—end-devices initiate their transmissions without any kind of

coordination [3]. LoRaWAN typically operates in license-free ISM bands in which the transmission power ($P_{\text{Tx}}$) and duty cycle are regulated. In Europe for instance, LoRaWAN networks mostly use sub-bands of the EU868 frequency band in which the limitations are typically $P_{\text{Tx}}$ of 14 dBm and a duty cycle of 1%. LoRaWAN is strongly uplink oriented but each uplink transmission is followed by two short receive windows for the reception of ACKs, downlink traffic or ADR commands (which can all be combined in the same packet). The ED might open additional receive windows if it operates in class B (Beacon) or class C (Continuously listening). We focus on class A (All EDs). Otherwise, the ED radio remains switched off, which greatly reduces energy consumption. LoRaWAN defines a set of LoRaMAC® commands to manage EDs over-the-air. In particular, these downlink commands allow to adapt the uplink transmission parameters such as $P_{\text{Tx}}$, *SF* and number of frame repetitions ($NB_{\text{Trans}}$). Many limitations of LoRaWAN in terms of scalability and effective throughput are inherent to ALOHA access [4,5]. Moreover, ensuring reliable uplink traffic handling by means of ARQ or any kind of feedback is challenging due to very limited downlink traffic capacity [5,6], even though improvements are possible [7]. Macro-diversity is a central feature of LoRaWAN: all GWs use the same frequency channels and each uplink frame is typically received and forwarded by several GWs, which also allows to estimate the ED's position, by measuring the time of arrival differences. We investigate the benefits of this redundancy and propose to take it into account to optimize the transmission parameters.

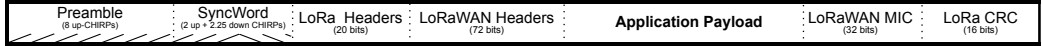

**Figure 2.** LoRaWAN uplink frame structure.

### 2.2. Metrics

The physical and MAC layers parameters *BW*, *SF*, *CR* and $NB_{\text{Trans}}$, allow for a trade-off between reliability and throughput. This multidimensional operational point is of great importance for LoRaWAN performance adjustment. Moreover, the communication reliability directly impacts the operational range of the transmission system. In our analysis we distinguish 3 types of losses, at different levels. The Frame Erasure Rate (*FER*) is the physical loss ratio between the ED and a given GW (i.e., without repetitions). The Packet Error Rate (*PER*) is the loss ratio between the ED and the NS. *PER* benefits from multiple gateways reception and frame repetition. The Data Error Rate (*DER*) is the loss ratio between the ED and the Application Server (AS), thus benefiting from the presence of an application layer inter-packet FEC algorithm, when available.

The throughput defines the transmission duration and thus the channel load over time which, in turn, determines in the overall system capacity. Hence, each of theses parameters which in their own way influence the transmission time spreading, is key to reduce the network air-time pressure, improve reliability and improve scalability.

Let us define the *ToA* per application bit, *ToA/b*, to be the overall time spend for the transmission of one application bit.

$$ToA/b = \frac{Time\text{-}On\text{-}Air}{Number\ of\ bits\ in\ the\ application\ payload.}$$

In Figure 3 we show the *ToA/b* cost to transmit a 25 bytes application payload with $SF \in [7..12]$, $CR \in \{\frac{4}{5}; \frac{4}{8}\}$ and $NB_{\text{Trans}} \in [1..3]$ with *BW*=125 kHz. The *ToA/b* cost smoothly increases as we move toward more robust transmission parameters. The transmission with the most robust configuration is 100 times more costly than the least expensive one.

Furthermore, the transmission duration strongly impact the ED energy consumption. It has even been shown that reducing the ToA is the main factor in reducing the Network Energy Consumption within a dynamic parameters allocation [8]. Thus, in the following we do not provide detailed results on energy consumption and focus on both the *ToA/b* and *DER*.

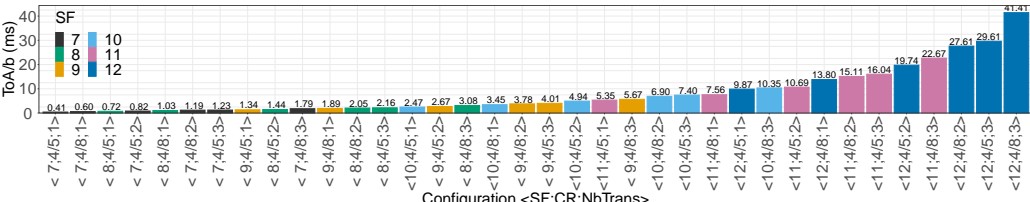

**Figure 3.** Time-On-Air by applicative bit (*ToA/b*) cost for 25 bytes applicative payload over a 125 kHz bandwidth for selected transmissions parameters.

### 2.3. Setup and Experiments

We have gathered an experimental dataset by recording LoRaWAN transmissions collected by several gateways in an urban area. We then replay the recorded frame series to assess the effect of adjusting various MAC parameters.

The test-bench consists of one ED (B-L072Z-LRWAN1 LoRa®/Sigfox™ Discovery kit.), placed indoors on the third floor of a residential building, sending traffic to the *The Thing Network* (TTN) community network through a set of gateways. The device transmits series of LoRaWAN frames and varies the transmission parameters from one frame to the next. We randomize the transmission parameters in order to avoid shadow correlations and moderate the effect of possibly congested frequency channels.

We present here the results from two measurements sessions. In both cases, we used three channels centered on 868.1, 868.3 and 868.5 MHz, with bandwidth $BW = 125$ kHz.

In the first session, we considered 48 combinations of ($P_{\text{Tx}}$, *SF*) value pairs. We set the LoRaWAN coding rate to the default $CR = \frac{4}{5}$ for intra-packet FEC. The payload was 15 Bytes resulting in LoRa frames with a Number of Symbols (*NS*) from 38 to 53. The experiment ran for a whole week and there were on average 4300 frames transmission attempts per series, that is, one frame every ≈2.4 min.

In the second experiment we extended the possible configurations to many more ($P_{\text{Tx}}$, *SF*, *CR*) combinations with a payload such that $48 \leq NS \leq 50$. We also extended the possible configurations for $SF = 7$ with $CR \in \{\frac{4}{5}; \frac{4}{6}; \frac{4}{7}; \frac{4}{8}\}$ and with 10 different frame sizes: $48 \leq NS \leq 298$. Notice that because each *SF* does not encode the same amount of bits and because the FEC add redundant Bytes to the payload in a non continuous manner, some value of NS might not be feasible with some transmissions parameters combination. Frame with up to 2 less symbols are then used. 336 transmission parameters configurations are thus tested. The experiment ran for twelves days and the dataset includes on average 940 frames transmissions attempts per series, that is, one frame every ≈ 20 min.

Table 1 summarizes the choice of experimental configurations.

**Table 1.** Transmissions configurations used in the experiments.

| | $P_{\text{Tx}}$ (dBm) | SF | CR | NS |
|---|---|---|---|---|
| **XP1** | $\{0; 2; 4; 6; 8; 10; 12; 14\}$ | [7..12] | $\frac{4}{5}$ | $\{38; 43; 48; 53\}$ (15 Bytes Payload) |
| **XP2** | $\{0; 2; 4; 6; 8; 10; 12; 14\}$ | 7 | $\{\frac{4}{5}; \frac{4}{6}; \frac{4}{7}; \frac{4}{8}\}$ | 50 |
| | | 7 | $\{\frac{4}{5}; \frac{4}{8}\}$ | $\{80; 98; 128; 178; 200; 224; 248; 280; 298\}$ |
| | | [8..12] | $\frac{4}{5}$ | 50 |

Twelve and thirteen TTN GWs showed up within the transmission range of the device during respectively the first and the second experiment. Among them, 8 were up for both experiments. Fifteen are GWs deployed in the Grenoble urban area within 5 km of the ED. Two GWs are outside the city at 6 km and 14 km from the ED with respectively 1000 m and 2000 m higher elevation from the city. We could not retrieve the position of one of the GWs. Figure 4 shows the position of the GWs in red and their distance from the ED (in yellow).

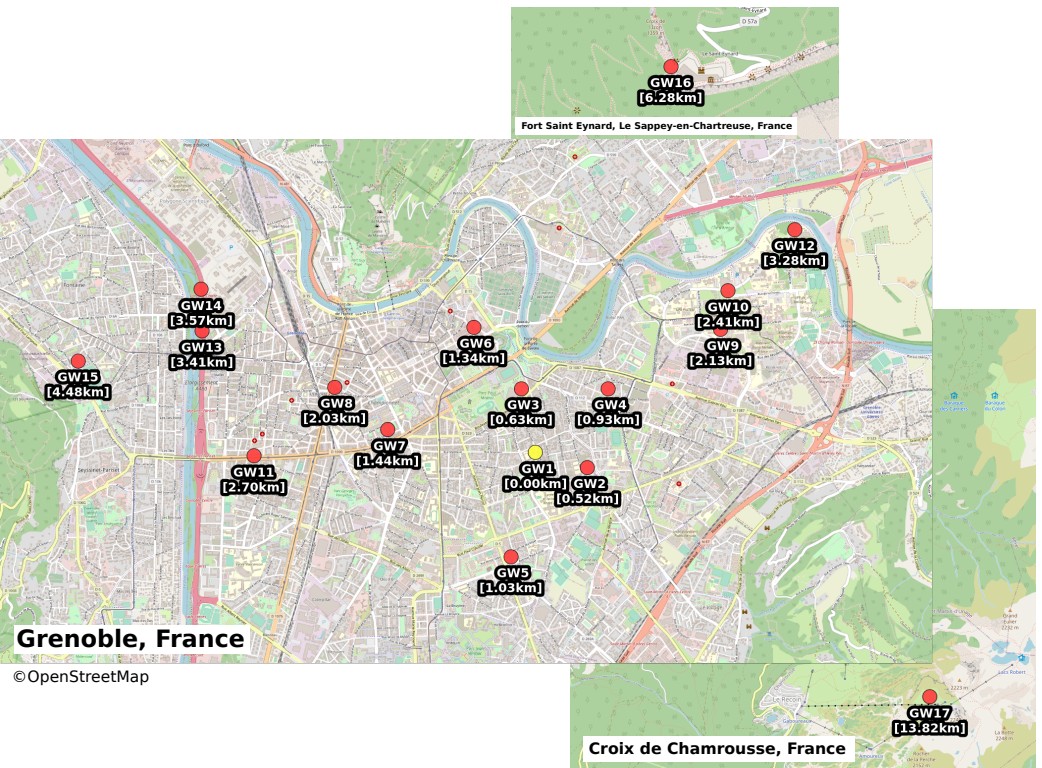

**Figure 4.** Experimental Setup.

We reject from the study the series with *FER* > 0.99. In these cases, the transmission is not robust enough and only residual frames are received. The valid data represents a total of 2319 independents LoRaWAN series of frames.

This set of measures captures the frame erasure patterns over a typical LoRaWAN urban network, and it is publicly available (https://gricad-gitlab.univ-grenoble-alpes.fr/coutaudu/lora-measurements, accessed on 29 March 2021).

## 3. Channel Characterization

The receive signal strength ($P_{Rx}$) and hence, the frame reception success, is determined by the transmission power, the antennas gains, and the Path Loss (PL, i.e., the radio signal attenuation between the two antennas). We decompose PL into three fading components: large scale fading (LSF), shadow fading (ShF) and temporal fading (TF).

LSF depends on the distance between the radios and the propagation medium path loss exponent $\gamma$. It may vary slowly in time with the evolution of the propagation medium. For instance the ambient air temperature and humidity vary with the climate condition and slightly modify $\gamma$.

ShF corresponds to obstructions over the main path, such as trees, buildings, walls, shutters, and so forth. LSF and ShF determine the average PL between a given ED and the GW. As a consequence, they determine if the ED is within communication range of the GW. LSF and ShF are often modeled using the log-distance path loss model : $PL(d) = PL_0 + 10\gamma \log_{10} \frac{d}{d_0} + \mathcal{X}$, where $PL_0$ is the path loss at the reference distance $d_0$ and $\mathcal{X}$ is a zero mean Gaussian random variable corresponding to the ShF, or using empirical models such as the Okumura-Hata models. We consider static EDs and GWs, so that the distance and obstructions on the radio path do not vary. So we consider that LSF and ShF are constant.

TF corresponds to the gain from multi-path propagation. It causes the PL to vary from one transmission attempt to the next, at least because the frequency channel changes randomly. TF determines the distribution of the PL around the average value; it has a major impact on the proportion of transmissions received above the demodulation threshold and on the power difference between colliding frames. Consequently, characterization of

the TF of the LoRa channel is key to studying and modeling reliability and scalability in LoRaWAN networks.

Below, we consider $P_{\mathrm{Rx}}$ with respect to the ambient noise, in terms of the Signal to Noise Ratio (*SNR*).

### 3.1. Rayleigh Channel Behavior

For all GWs, our experiments show that the *SNR* distribution roughly follows a Rayleigh channel exponential distribution. This distribution is expected in our setup in which there is no line-of-sight and thus the propagation is likely to be highly multipath, with no dominant path. The *SNR* over a Rayleigh channel follows an exponential distribution with cumulative distribution function $CDF_{\mathrm{exp}}(x) = 1 - e^{-x}$ (and its reciprocal $CDF_{\mathrm{exp}}^{-1}(x) = -\log(1-x)$), multiplied by a factor $\overline{SNR}$ corresponding to the *SNR* mean, that is, the gain factor from the unit mean exponential distribution (UMED). Figure 5 shows the *SNR* distribution for several GWs and several $P_{\mathrm{Tx}}$. Theoretical *SNR* distributions over a Rayleigh channel appears in red in Figure 5, with manually fitted gain shifts. The histogram does not follow perfectly the exponential distribution because as we reduce $P_{\mathrm{Tx}}$, the *SNR* distribution translates to the left, towards lower values: thus, more and more frames fail to meet the GW sensitivity when the $P_{\mathrm{Rx}}$ is below demodulation floor *Df*. The sensitivity for each considered SF is marked by an arrow in Figure 5. Below this point, most of the frames are lost, resulting in a progressively more and more censored sample as $P_{\mathrm{Tx}}$ decreases. It is important to note that because of this censoring of the lower *SNR* frames, the $\overline{SNR}$ is biased compared to the one which would include the lost frames.

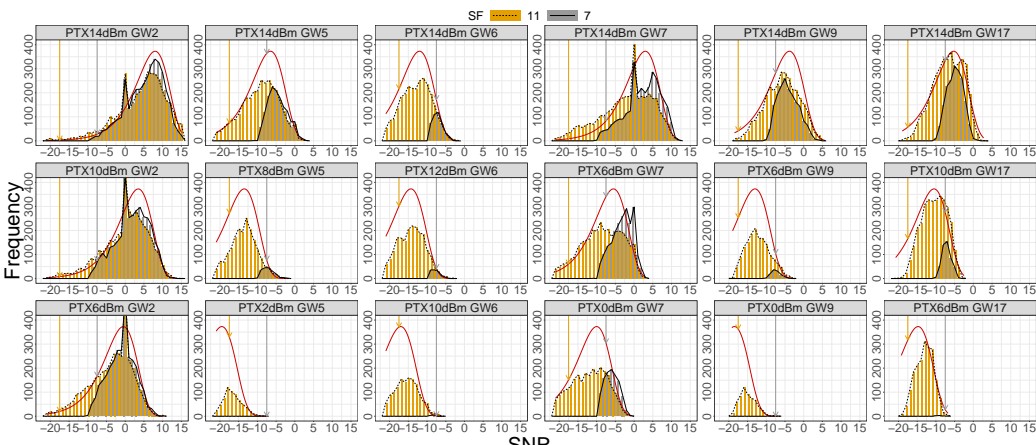

**Figure 5.** Distribution of the measured *SNR* of several LoRaWAN series of frames with *SF* 11 and 7, compared to an exponential distribution curve in red (manually centered), for several $P_{\mathrm{Tx}}$. Yellow and black arrows mark each *SF* 11 and 7 demodulation floor (typical values from the documentation [1]).

Besides, notice that there is an artifact at 0 dB due to a bad interpretation of some frames by the hardware monitoring system, which wrongly marks them with a 0 value and it is not possible to distinguish them from the frames received with an actual 0 dB SNR.

Assuming that the channel is Rayleigh, and that meeting *Df* is necessary and sufficient for successful reception in absence of collision, *FER* is:

$$FER = CDF_{\mathrm{exp}}\left(10^{\left(\frac{Df - \overline{SNR}}{10}\right)}\right), \tag{1}$$

with $\overline{SNR}$, the SNR mean and *Df*, the demodulation floor, both in dB. Obviously, any of these three parameters can be obtained from the two others:

$$Df - \overline{SNR} = \left(10 \times \log_{10}(-\ln(1 - FER))\right). \tag{2}$$

These relations capture the erasures from fading but collisions in a loaded network would introduce a bias.

### 3.2. Impacts of LoRa Intra-Frame Forward Error Correction

LoRaWAN defines an intra-frame Forward Error Correction scheme derived from an Hamming erasure correction code. A couple of studies using reverse engineering and analysis offer a glimpse of the algorithm and its performance [9,10]. These studies predict error detection capabilities only for $CR \in \{\frac{4}{5}, \frac{4}{6}\}$ and then more and more error correction for $CR \in \{\frac{4}{7}, \frac{4}{8}\}$. As shown in Figure 6 for relevant series of frames, our measurements do not corroborate these findings as we already observe a reliability gain for $CR = \frac{4}{6}$.

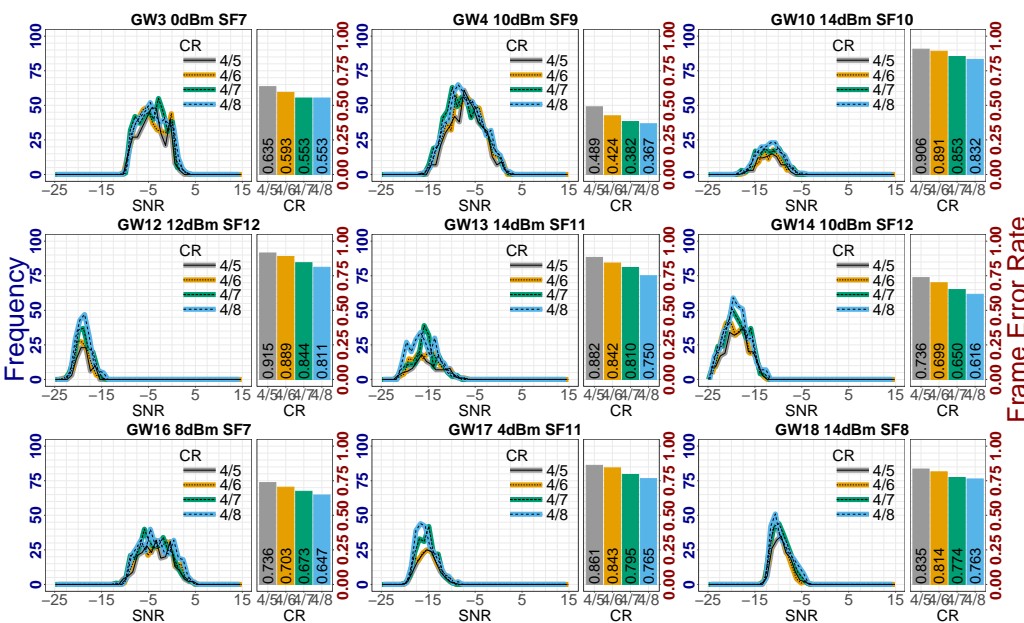

**Figure 6.** Distribution of the measured *SNR* and *FER* as a function of the *CR* for selected series of frames.

To quantify this reliability gain, we find the demodulation floor difference $\Delta Df$ between a reference configuration ($CR = \frac{4}{5}$) and other configurations ($CR \in \{\frac{4}{6}, \frac{4}{7}, \frac{4}{8}\}$), all other transmissions parameters (*SF*, $P_{Tx}$, *NS*) being equal. Assuming that all losses are caused by the channel gain variability due to fast fading (and that any frame received below *Df* is lost), we estimate $\left(Df - \overline{SNR}\right)_{\text{ref}}$ the margin between *Df* and $\overline{SNR}$ for the reference configuration by applying Equation (2) to $FER_{\text{ref}}$.

$$\left(Df - \overline{SNR}\right)_{\text{ref}} = \left(10 \times \log_{10}(-\ln(1 - FER_{\text{ref}}))\right) \tag{3}$$

Using Equation (1) we can calculate the estimated $\widehat{FER}$ obtained by considering that *Df* is improved (i.e., shifted to the left) by $\Delta Df$ (in dB).

$$\widehat{FER} = CDF_{\exp}(10^{\left(\frac{\left(Df - \overline{SNR}\right)_{\text{ref}} + \Delta Df}{10}\right)}) \tag{4}$$

Finally, we find the $\Delta Df$ value for which $\widehat{FER}$ fits the experimental *FER*, $FER_{XP}$. We find the adequate $\Delta Df$ value using the Ordinary Least Square (OLS) method which consists in minimizing $\chi^2(\Delta Df) = \sum \left(\widehat{FER} - FER_{XP}\right)^2$.

Figure 7 shows $FER_{\text{ref}}$ and $FER_{XP}$ for each *CR*. The black curves show the fitted reliability gain $\Delta \widehat{FER} = FER_{\text{ref}} - FER_{XP}$ for the fitted $\Delta Df$. The experimental *FER* gain $\Delta FER_{XP}$ distributions match the shape of the theoretical *FER* gain distributions induced by sensitivity gains $-0.40$ dB, $-0.88$ dB, $-1.21$ dB for respectively *CR* $\frac{4}{6}$, $\frac{4}{7}$ and $\frac{4}{8}$.

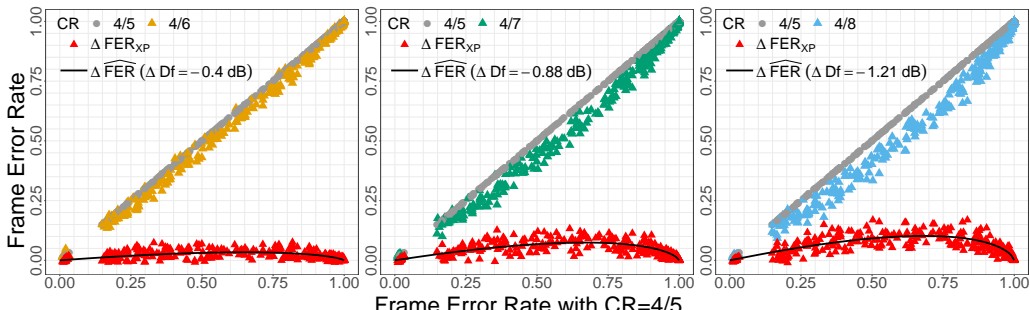

**Figure 7.** Comparison of the Frame Error Rate with $CR = \frac{4}{5}$ against $CR = \frac{4}{6}$, $CR = \frac{4}{7}$ and $CR = \frac{4}{8}$. The black curve is the computed *FER* gain expected for sensitivity gain over a Rayleigh channel.

### 3.3. Impact of the Frame Length

Frame length is expected to have a minor impact on successful reception because the latter depends on successful initial synchronization [11]. Still, our experimental measurements show a noticeable impact of frame length on the *FER*. To assess the magnitude of this effect, we gather measurements for diverse frame lengths using *SF*7, in Figure 8. It shows the *SNR* distribution for frames of various lengths $NS \in \{48, 128, 298\}$ and the *FER* for various values of $NS \in [48..298]$. In particular 48 and 298 that are key values to compare precisely between different *SF* and *CR*.

Reliability clearly drops as the number of symbols per packet increases. Up to 20 percentage points of *FER* can be lost between the shortest and longest frames.

In Figure 9, we compare again the experimental *FER* difference $\Delta FER_{\text{Exp}}$ and the expected *FER* difference $\widehat{\Delta FER}(\Delta Df)$. using the same methodology as in Section 3.2, using the configuration with $NS = 298$ as reference. The OLS fit results are given in Table 2. With our experimental setup, reducing frame length from $NS \in [296..298]$ to $NS = 128$ and $NS = 48$ produces a sensitivity gain of respectively $-1.28$ dB and $-2.5$ dB.

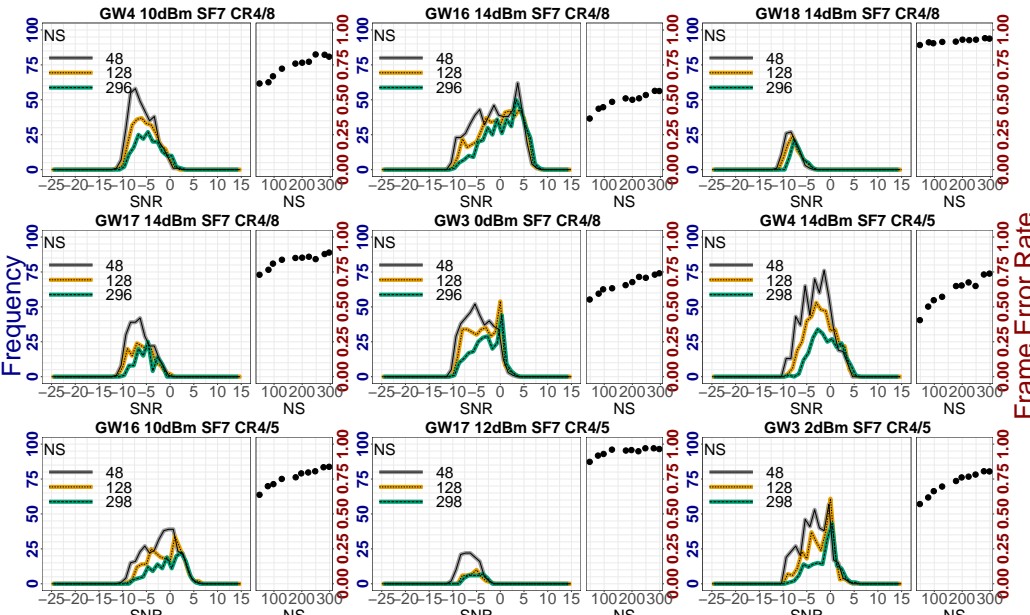

**Figure 8.** Distribution of the measured *SNR* and *FER* as a function of the number of symbols (*NS*) for selected series of frames.

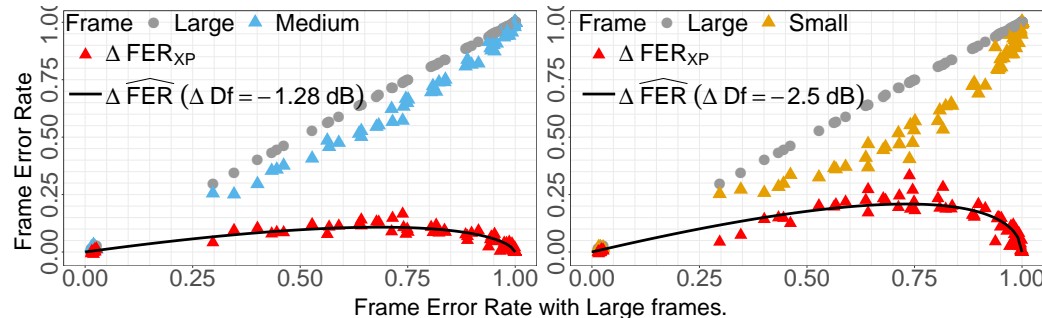

**Figure 9.** Comparison of the Frame Error Rate with $NS = 48$ against $NS = 128$ and $NS \in [296..298]$. The black curve is the computed *FER* gain expected for sensitivity gain over a Rayleigh channel.

**Table 2.** Ordinary Least Square (OLS) fitted values of the $\Delta Df$ gains for various frame lengths compared to $NS \in [296..298]$.

| NS | 48 | [78..80] | [96..98] | 128 | [176..178] | 200 | [223..224] | 248 | [278..280] | [296..298] |
|---|---|---|---|---|---|---|---|---|---|---|
| $\Delta Df$ (dB) | −2.50 | −1.93 | −1.74 | −1.28 | −0.85 | −0.70 | −0.48 | −0.39 | −0.11 | 0 |
| $\chi^2$ ($\times 10^{-4}$) | 8.844 | 4.903 | 3.673 | 2.366 | 2.018 | 1.693 | 1.393 | 2.056 | 0.949 | 0 |
| Frame Length Ratio | 0.16 | 0.27 | 0.33 | 0.43 | 0.6 | 0.96 | 0.75 | 0.83 | 0.94 | 1 |

However, one must keep in mind that the sensitivity gain from a reduced physical frame length comes as a relative expense of the preamble and protocol header length. When the application data payload shrinks, most of the *ToA* consists in the physical preamble and protocol header. For instance, with $SF = 7$, $CR = \frac{4}{5}$ and $BW = 125$ kHz, a 50 symbols frame encapsulates 13 application data bytes with $TOA/b = 0.59$ ms, whereas a 298 symbols frame encapsulates 188 application data bytes with $TOA/b = 0.21$ ms, that is, 2.8 times lower. We illustrate this in Figure 10 the *ToA/b* against the application data payload length from 1 to 250 bytes. Notice that the effective *ToA/b* for 30 bytes application data payload at *SF* 7 is equivalent to the one for 250 bytes application data payload at *SF* 8.

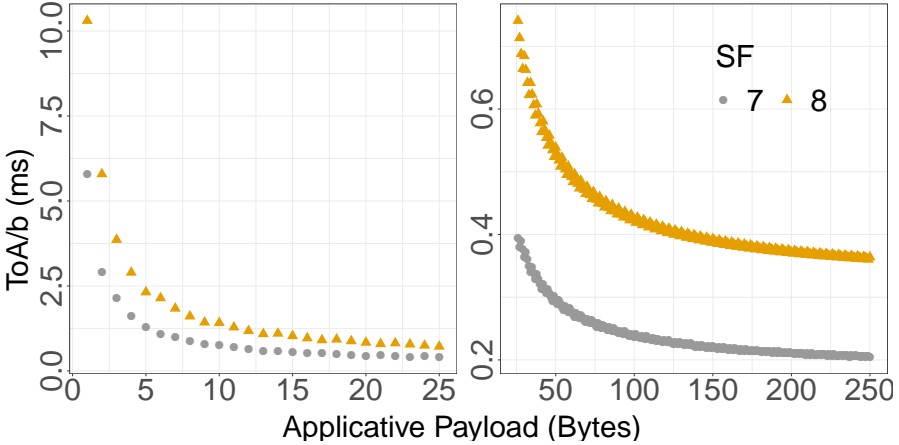

**Figure 10.** *ToA/b* as a function of the application data payload length for *SF* 7 and 8 with *BW* = 125 kHz and $CR = \frac{4}{5}$.

### 3.4. Channel Burstiness Behavior

Another crucial point from the perspective of providing reliable communication is erasures burstiness. We compare in Figure 11 the proportion of frames lost in erasure bursts in a simulated channel with independent and identically distributed (iid) losses vs. in our experimental data. The experimental data erasure patterns are close enough to be approximated in the following as an iid erasure channel.

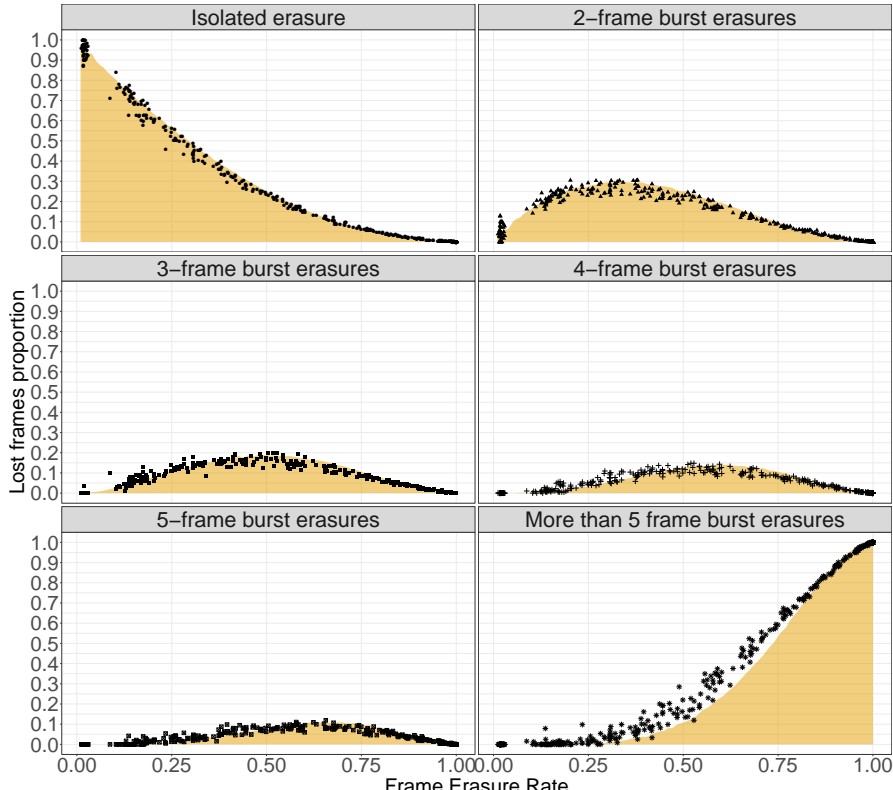

**Figure 11.** Dots marks the experimental proportion of frames lost in bursts of various sizes. The colored areas correspond to a simulated independent and identically distributed (iid) channel.

## 4. Transmissions Parameters Selection

### 4.1. $ADR_{opt}$ a Configurable and Optimized ADR

#### 4.1.1. $ADR_{opt}$ Algorithm

Based on the insight gained from our experimental measurements, we propose an NS-side optimized ADR algorithm $ADR_{opt}$, detailed in Algorithm 1 which dynamically adapts the transmission parameters to get the most out of the available radio links. For the sake of simplicity we restrain the available Tx configurations to any combinations of $SF \in \{[7..12]\}$, $CR = \frac{4}{5}$ and $NB_{Trans} \in [1..3]$ over 125 kHz channels at 868.3, 868.5 and 868.8 MHz. Also, we do not take into account the sensitivity variation due to the frames size.

In the following, we consider a constant $P_{Tx}$, as $P_{Tx}$ is reduced from its maximal value only when the signal is very strong and both $SF$ and $NB_{Trans}$ are set to the lowest values, so that $P_{Tx}$ has little influence on the performance of ADR in terms of $PER$ and $ToA$.

$ADR_{opt}$ extrapolates a presumable $PER$ for each $[SF; NB_{Trans}]$ pair from the observation on the channel over the previous transmission period. $ADR_{opt}$ then chooses the transmission parameters to maintain $PER$ lower than the target level $PER_{target}$.

The $ADR_{opt}$ $FER$ estimation function is based on the assumption that the channel is Rayleigh as described in Section 3.1. In the following, we refer to $\widehat{SNR}$ as the computed extrapolation, therefore imperfect, of $\overline{SNR}$. For a given GW, we can compute $\widehat{SNR}$ over a period of time. Assuming that the channel does not change drastically for the next period, we can compute the expected $FER$ for any transmission parameters. And eventually, with the assumption that each transmission and reception by different GWs are independent, we can extrapolate the $PER$ by combining the $FER$ for each of the GWs in the reception range of the transmitter.

**Algorithm 1** $ADR_{\text{opt}}$-Server algorithm.

---

1: ChHistory(20) // Initialization of the list of the last 20 frames received.

2: $PER_{\text{target}}$

3: **while** true **do**

4:     ACK_Req=waitRx();

5:     **if** (ACK_Req) **then**

6:         // Compute a prediction of the *PER* for each configuration.

7:         **for all** $GW \in$ receptionGW(*ChHistory*) **do**

8:             **for** $SF \in \{7; 8; 9; 10; 11; 12\}$ **do**

9:                 $FER =$ estimateFer($GW, SF, ChHistory$)

10:                 **for** $NB_{\text{Trans}} \in \{1; 2; 3\}$ **do**

11:                     $PER_{\text{predic}}[SF; NB_{\text{Trans}}] \ast = FER^{NB_{\text{Trans}}}$;

12:                 **end for**

13:             **end for**

14:         **end for**

15:         $PER_{\text{local target}} = PER_{\text{target}}$;

16:         $PER_{\text{current}} =$ getPER(*ChHistory*);

17:         **if** $PER_{\text{current}} > PER_{\text{target}}$ **then**

18:             *// FEC may fail to recover all the lost frames thus $PER_{\text{local target}}$ is reduced to better compensate erasures and achieve recovery.*

19:             $PER_{\text{local target}} = \max(0.01, PER_{\text{target}} - (PER_{\text{current}} - PER_{\text{target}})$;

20:         **end if**

21:         *//Choose the best configuration that fits the PER requirement and minimal ToA.*

22:         setValidLowestToAConfig($PER_{\text{predic}}, PER_{\text{local target}}$)

23:     **end if**

24: **end while**

---

Because a loaded network would distort the estimation of $\widehat{SNR}$ from the observed *FER*, we choose to rely on another characteristic of the exponential distribution that would not suffer such bias: the highest received *SNR*, $SNR_{\max}$. As the channel history buffer keeps a limited number of received frames (the TTN NS keeps only the last 20 frames.), we have to compute what would be the size of the sample $S$ with its censored part, that is, the erased frames:

$$size_S = \frac{20}{(1 - PER_{\text{current}})} \times NB_{\text{Trans}}$$

We then estimate what would be the maximal value of such a sample following UMED. We approximate the theoretical $SNR_{\text{Max}}(S_{\text{UMED}})$ by $SNR_{\approx\text{Max}}(S_{\text{UMED}})$ that we define as the middle of the interval in which there is 90% chances that this maximum *SNR* lies, with $size_S$ trials:

$$SNR_{\approx\text{Max}}(S_{\text{UMED}}) = \frac{\left(10 \times \log_{10}\left(CDF_{\exp}^{-1}\left(0.95^{(1/size_S)}\right)\right)\right)}{2}$$
$$+ \frac{\left(10 \times \log_{10}\left(CDF_{\exp}^{-1}\left(0.05^{(1/size_S)}\right)\right)\right)}{2}.$$

From this we estimate $\widehat{SNR}$ in dB:

$$\widehat{SNR} = ChHistory_{GW}(SNR_{\max}) - SNR_{\approx \text{Max}}(S_{\text{UMED}}) \, .$$

We combine this $\widehat{SNR}$ with the typical SNR demodulation floor of LoRa [1]:

$$SNR_{\text{floor}<SF>} = (-20) + ((12 - SF) * 2.5).$$

Thus the *FER* is:

$$FER_{<GW_i;SF>} = CDF_{exp}(10^{\left(\frac{SNR_{\text{floor}<SF>} - \widehat{SNR}}{10}\right)}).$$

Which leads to:

$$PER_{<NB_{\text{Trans}};SF>} \approx \prod_{\forall GW_i} (FER_{<GW_i;SF>})^{NB_{\text{Trans}}}$$

These formulae compute an accurate approximation of the *FER* and consequently the *PER* for each available transmission parameters.

With $ADR_{\text{opt}}$ the $PER_{\text{target}}$ is an input parameter of the algorithm that can be fixed to an arbitrary value. Thus $ADR_{\text{opt}}$ adapts to arbitrary reliability needs.

### 4.1.2. $ADR_{\text{opt}}$ Performance Simulation

We assume a perfect downlink channel which allows to transmit all the $ADR_{\text{opt}}$ piggybacked commands and parameters into downlink ACKs. The Rayleigh channel describes a series of frames with a fixed *SNR* mean ($\overline{SNR}$), which corresponds to fixed positions of the node and the gateway. For each frame $f$, $SNR_f = \overline{SNR} \times X$ where $X$ is a random variable following the UMED distribution function. Thus, a frame is dropped if $SNR_f < SNR_{\text{floor}<SF>}$. We simulate this for $\overline{SNR}$ in $[-30..10]_{dB}$ by steps of 0.5 dB with series of 6000 frames repeated 60 times.

$ADR_{\text{opt}}$ performance over the simulated Rayleigh channel appears in Figures 12 and 13 in presence of 1, 2, 4 and 8 GWs when the $\overline{SNR}$ to all GWs are equal. Notice that in a configuration with unequal $\overline{SNR}$, GWs with relatively low $\overline{SNR}$ bring little benefits: the overall performances tends to be the performances of a network with only the best $\overline{SNR}$ GW, that is, most of the time the closest one.

$ADR_{\text{opt}}$ sharply adapts the transmission parameters to reaches $PER_{\text{target}}$. We distinct three cases:

- $\overline{SNR}$ is too low and $PER_{\text{target}}$ cannot be met. In this case, $ADR_{\text{opt}}$ uses the most robust and most TOA expensive transmission configuration available. The ability to meet the required $PER_{\text{target}}$ is conditioned by the most robust available configuration. The most robust parameter combination is, in our case, *SF* 12 with $NB_{\text{Trans}} = 3$. It is also conditioned by the number of GWs in range.
- $\overline{SNR}$ is medium and *PER* is just above $PER_{\text{target}}$. In this case, $ADR_{\text{opt}}$ uses the transmission configuration corresponding to the smallest ToA while meeting the reliability requirement. There are *PER* fluctuations due to the discrete nature of available configurations and their corresponding reliability. These fluctuations therefore also depend on the slope of the $CDF_{\text{exp}}$ at the targeted focal point $PER_{\text{target}}$. These fluctuations are accentuated by the number of gateways.
- $\overline{SNR}$ is high and *PER* is far below $PER_{\text{target}}$. In this case, $ADR_{\text{opt}}$ uses the least TOA-intensive transmission configuration, *SF* 7 with $NB_{\text{Trans}} = 1$, and this end-device will over perform in terms of reliability.

Figures 12 and 13 also compare $ADR_{\text{opt}}$ with the TTN ADR default implementation ($ADR_{\text{TTN}}$). $ADR_{\text{TTN}}$ reduces the *SF* whenever ($ChHistory_{GW}(SNR_{\max}) - SNR_{\text{floor}<SF>}$) > *margin* with a default margin of 15 dB. Besides, $ADR_{\text{TTN}}$ increases and decreases $NB_{\text{Trans}}$, with a ceiling at $NB_{\text{Trans}} = 3$, whenever *FER* < 0.7 and *FER* > 0.9, respectively. $ADR_{\text{TTN}}$

relies on the EDs loss of connectivity denoted by the lack of downlinks (96 by default) to increase the *SF* and thus regain connectivity. More details of the $ADR_{TTN}$ algorithm can be found in previous works [12,13].

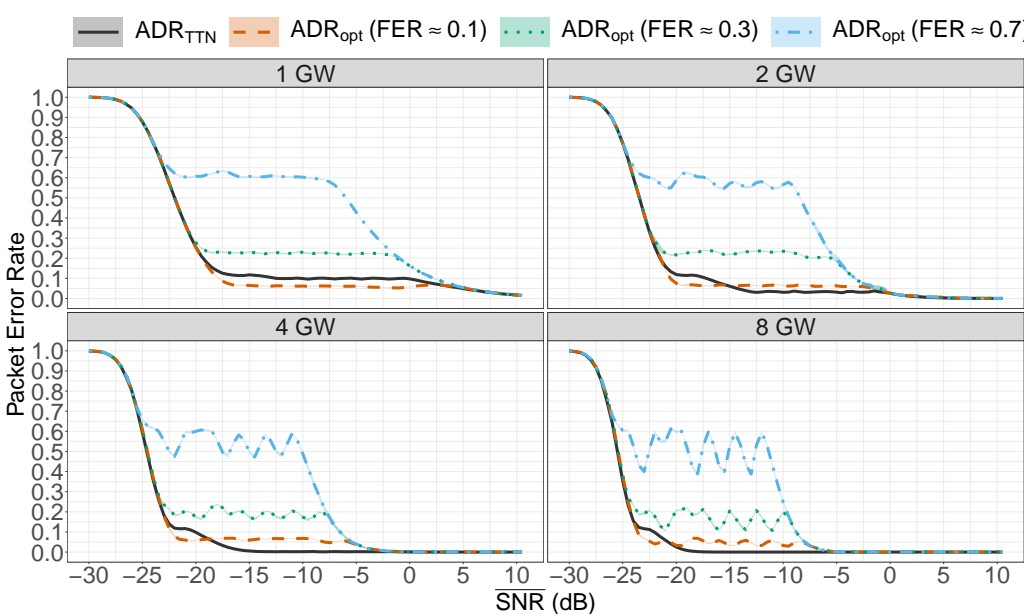

**Figure 12.** *DER* as a function of $\overline{SNR}$ for the simulated series of frames with multiple GWs (99% confidences interval plots).

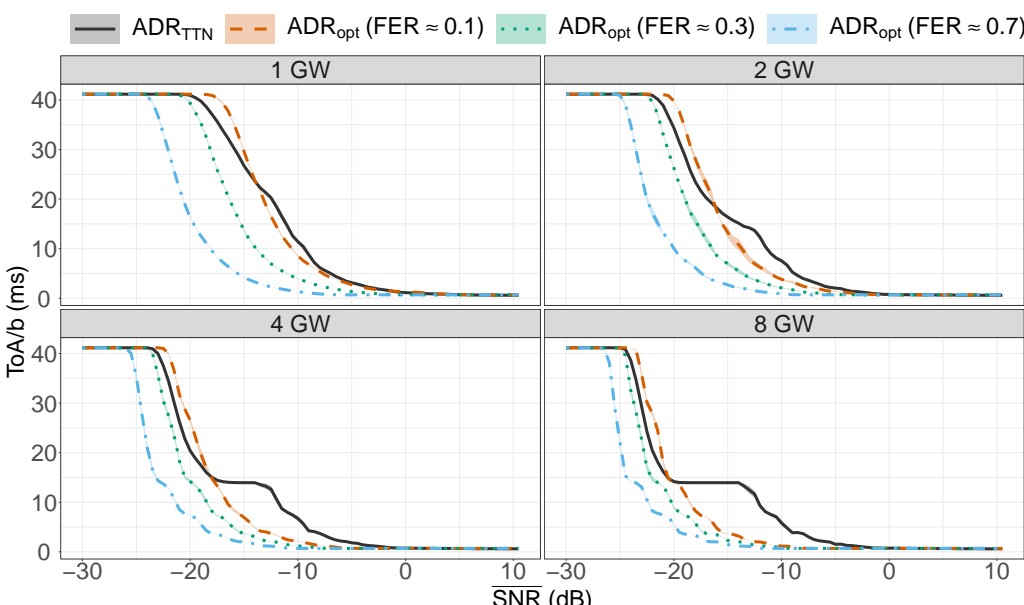

**Figure 13.** *ToA/b* as a function of $\overline{SNR}$ for the simulated series of frames with multiple GWs (99% confidences interval plots).

### 4.1.3. $ADR_{opt}$ Performance on Replayed Traces

We ran the experiments over several subsets of our real world transmission records. It appears that the reachable GWs can be classified following their *SNR* range. Figures 14 and 15 show the results for these subsets: GWs 5 and 6 that have low *SNR* (respectively $\overline{SNR} \approx -8.1$ dB and $\overline{SNR} \approx -12.1$ dB with $P_{Tx} = 14$ dBm), GWs 9 and 17 that have medium *SNR* (respectively $\overline{SNR} \approx -5.8$ dB and $\overline{SNR} \approx -6.6$ dB with $P_{Tx} = 14$ dBm), GWs 2 and

7 that have high *SNR* (respectively $\overline{SNR} \approx 4.6\,\text{dB}$ and $\overline{SNR} \approx -0.4\,\text{dB}$ with $P_{\text{Tx}} = 14\,\text{dBm}$), and finally the aggregation of GW 2, 5, 6, 7, 9 and 17.

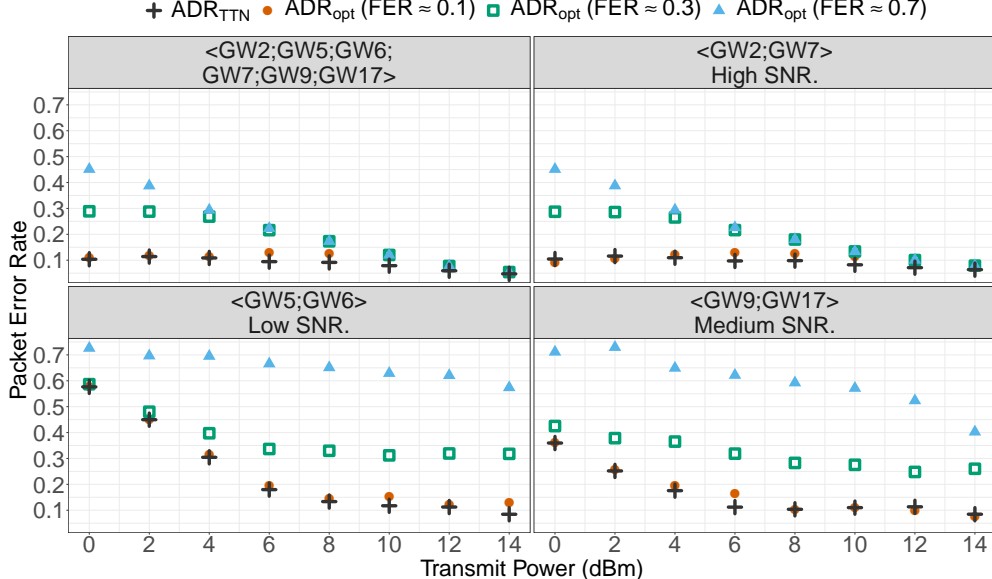

**Figure 14.** *DER* as a function of $P_{\text{Tx}}$, for selected real world series of frames replays.

The results derived from our real world transmission traces confirm the simulations of Section 4.1.2. For any subset and $P_{\text{Tx}}$ configuration, *ADR*opt provides adequate tuning for the transmissions and either $PER < PER_{\text{target}}$ is achieved or the most robust available configuration is used. Notice that the performances for the subset with GWs 2, 3, 4, 5, 6 and 8 is strongly dominated by the GWs providing the best signal reception, that is, GWs 2 and 3. As a consequence, its performances is just slightly better than the subset with GWs 2 and 3.

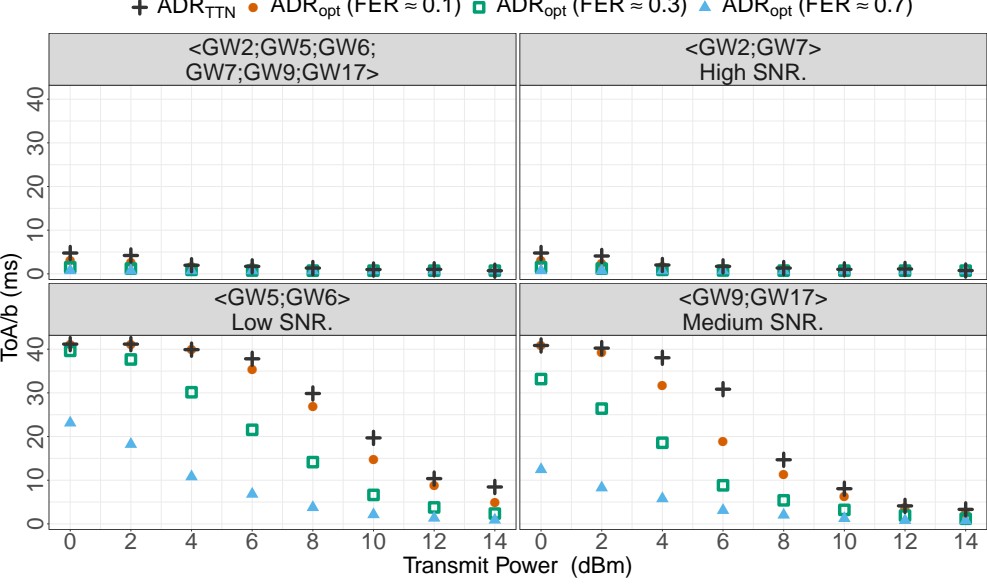

**Figure 15.** *ToA/b* as a function of $P_{\text{Tx}}$, for selected real world series of frames replays.

## 4.2. *ADR*IFECC *for High Reliability*

### 4.2.1. Inter-Frame Forward Error Correction for LoRaWAN

The finding that the channel can be modeled as a Rayleigh channel as detailed in Section 3.1 is key to understand the challenge to efficiently provide high reliability

($DER < 0.01$) in LoRaWAN. The fact that the distribution of the frames received power follows an exponential distribution, implies that a fraction of the frames faces a significantly higher path loss than the average. For instance, 10% of the transmissions will face a $-9.8$ dB $SNR$ from the $\overline{SNR}$. Likewise, respectively 5% and 2% of the frames will face a $-12.9$ dB and $-16.9$ dB $SNR$ from the $\overline{SNR}$. As a consequence, either an extremely robust transmission configuration, therefore over-robust most of the time, is used to received those few "very low SNR" frames, either the communication faces this "erasure floor". The solutions based on acknowledgement frames, such as Automatic Repeat reQuest (ARQ), are strongly limited by the asymmetry of LoRaWAN networks where downlink transmission opportunities are scarce [6]. Channel coding solutions based on Inter-Frames Erasure Correction Code (IFECC) recover the losses by introducing redundancy with time diversity in the communication and can provide high reliability in LoRaWAN [14,15].

The systematic repetition implemented by the LoRaWAN MAC layer parameter $NB_{\text{Trans}}$, is the simplest kind of IFECC. The redundancy is a duplicate of the original frame, transmitted successively, thus it is weak against burst losses. Even though, as stated in Section 3.4, the channel erasures are close to iid, the probability to loose multiple frames in a row mechanically increases with the $FER$. For instance, over a $FER = 0.25$ iid erasures channel around 15% of the lost frames are lost in burst of length $\geq 3$. By replaying our experimental data set, we emulated the MAC layer parameter $NB_{\text{Trans}}$ over our experimental data and the results are illustrated in Figure 16 with the $PER$ as a function of the $FER$ for various $NB_{\text{Trans}}$. Repetition IFECC indeed provides important reliability improvement: $NB_{\text{Trans}} = 2$ reduces $PER$ by more than 20 percentage points over a channel with $FER = 0.5$. Increasing $NB_{\text{Trans}}$ from 2 to 4 reduces $PER$ by around 20 percentage points over a channel with $FER = 0.65$. $NB_{\text{Trans}}$ can be increased to reduce the $PER$ without changing the $SF$ and with no decoding latency. However, even with $NB_{\text{Trans}}$ as high as 32, residual erasures appear over a channel with $FER > 0.7$. As expected, with the $FER$ increase, systematic repetition quickly fails to provide high reliability as it is weak against burst erasures and residual erasures are left uncovered. Moreover it becomes impractical due to its tremendous channel occupation overhead.

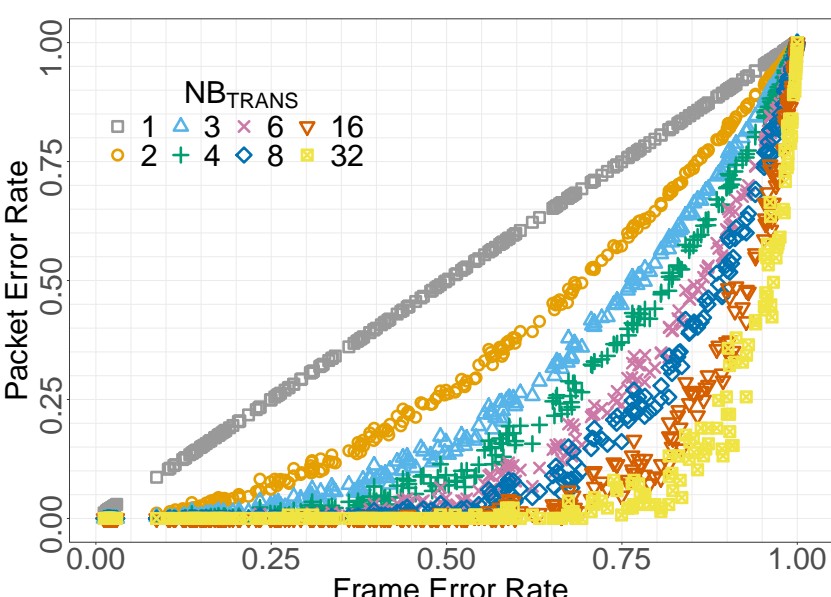

**Figure 16.** Experimental *PER* against *FER* for several $NB_{\text{Trans}}$.

In the following, we use an IFECC scheme based on a linear combination of packets, LoRaFFEC [16], which is efficient to recover the typical residual errors from a Rayleigh channel up to $FER \approx 0.3$. LoRaFFEC spreads the redundancy over many frames and so, introduces more time diversity. In practice, LoRaFFEC is computed and spread over 128 frames. Notably, the redundancy spreading is higher than the default ADR downlink

transmission period, limited to ACK_LIMIT + ACK_DELAY = 96 frames. Moreover, LoRaFFEC does not require additional downlink signaling. The redundant data overhead of LoRaFFEC, can either be transmitted in separate frames or piggybacked into existing ones. Here, we choose to piggyback, as the LoRa and LoRaWAN headers overhead are then paid only once, which improves the *ToA/b*. The frame are lengthened by a ratio $< 2$: A 15 bytes application payload needs 53 symbols at *SF* 7. This becomes with LoRaFFEC $1 + (15 + 3) \times 2 = 37$ bytes, and 83 symbols at *SF* 7 (with $CR = \frac{4}{5}$). So, the sensitivity loss is not prohibitive as it stays less than one dB as detailed in Section 3.3.

We define $ADR_{IFECC}$ the combination of the IFECC with $ADR_{opt}$. $ADR_{opt}$ is set to keep the *PER* above the correction threshold of the IFECC, that is, *PER* > 0.3 in our case. The IFECC recovers the remaining erasures and provides high reliability with *DER* < 0.01.

### 4.2.2. $ADR_{IFECC}$ Performance Simulation

We compare the performances of $ADR_{IFECC}$ with the default ADR implementation of TTN ($ADR_{TTN}$). We consider 15 bytes applicative payload. For the sake of simplicity we do not take into account the sensitivity impact of the varying frame length. The simulated channel is the same as in Section 4.1.2.

$ADR_{IFECC}$ performance over the simulated Rayleigh channel appears in Figures 17 and 18 in presence of 1, 2, 4 and 8 GWs when the $\overline{SNR}$ to all GWs are equal. $ADR_{IFECC}$ sharply adapts the transmission parameters to reaches *DER* < 0.0.1. For instance, in Figure 18 $ADR_{IFECC}$ provides *DER* < 0.01 over a single GW network with $\overline{SNR} \geq -21.5$ dB. This threshold is reduced as the number of GWs increases. $ADR_{IFECC}$ provides *DER* < 0.01 over an 8 GWs network with $\overline{SNR} \geq -25$ dB to all GWs.

However, as shown in Figure 18, $ADR_{IFECC}$ ToA is higher than $ADR_{TTN}$ for channels with low $\overline{SNR}$ ($-17$ dB and $-23$ dB for respectively 1 or 8 GWs). This corresponds to the extra energy invested by $ADR_{IFECC}$ to achieve a more reliable communication than $ADR_{TTN}$. For better $\overline{SNR}$ values, the transmissions parameters adjustments of $ADR_{IFECC}$ are more fine-grained and the same reliability is obtained for lower ToA as shown in Figure 18.

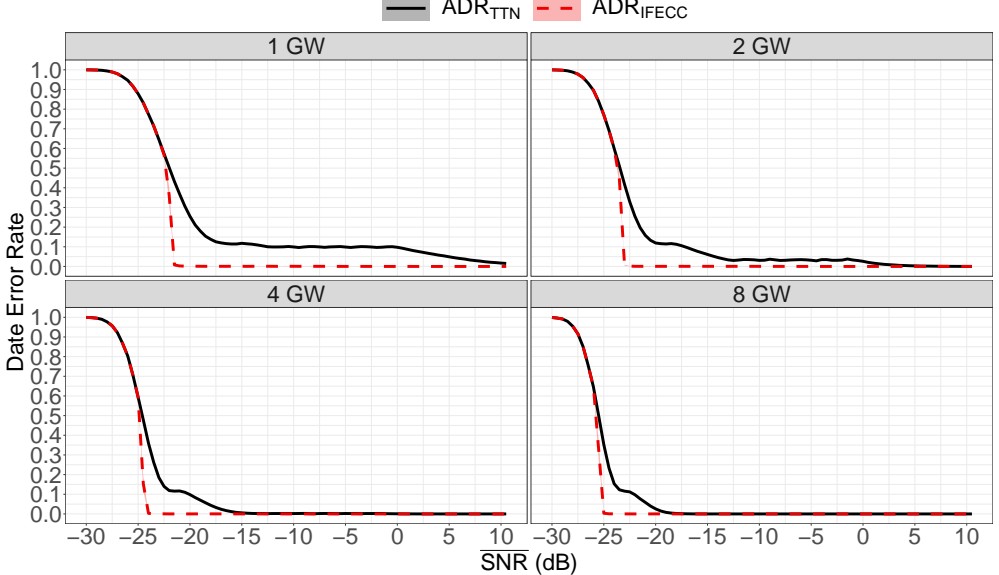

**Figure 17.** *DER* against $\overline{SNR}$ for the simulated series of frames with a yellow dashed line to mark the 0.01 threshold (99% confidences interval plots).

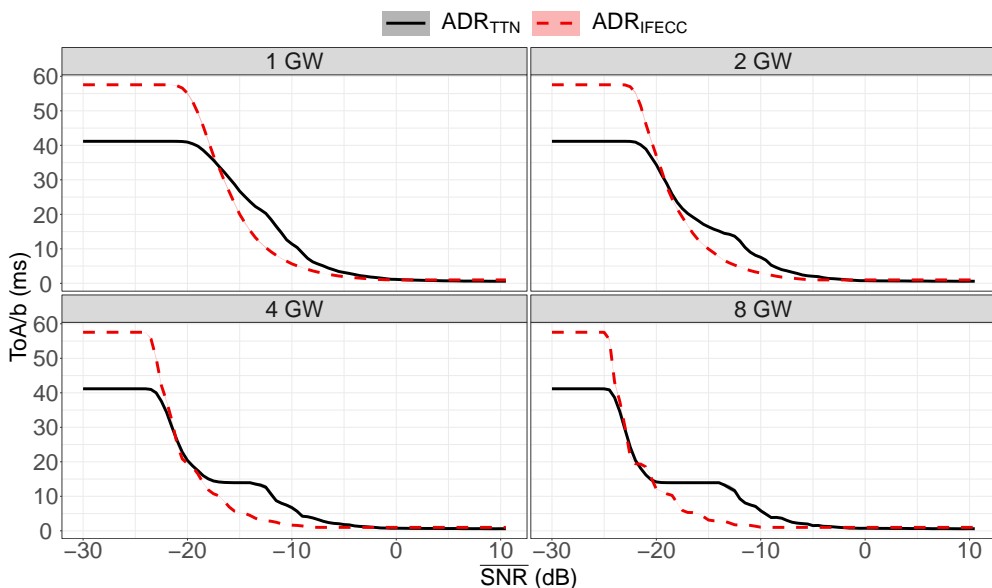

**Figure 18.** *ToA/b* as a function of $\overline{SNR}$ for the simulated series of frames with several GWs (99% confidences interval plots).

### 4.2.3. $ADR_{IFECC}$ Performance on Replayed Traces

The results from real world traces replay, with same subset as in Section 4.1.3 shown in Figures 19 and 20 confirm the simulation results. Either the most robust transmission parameters are used or $DER < 0.01$ is achieved.

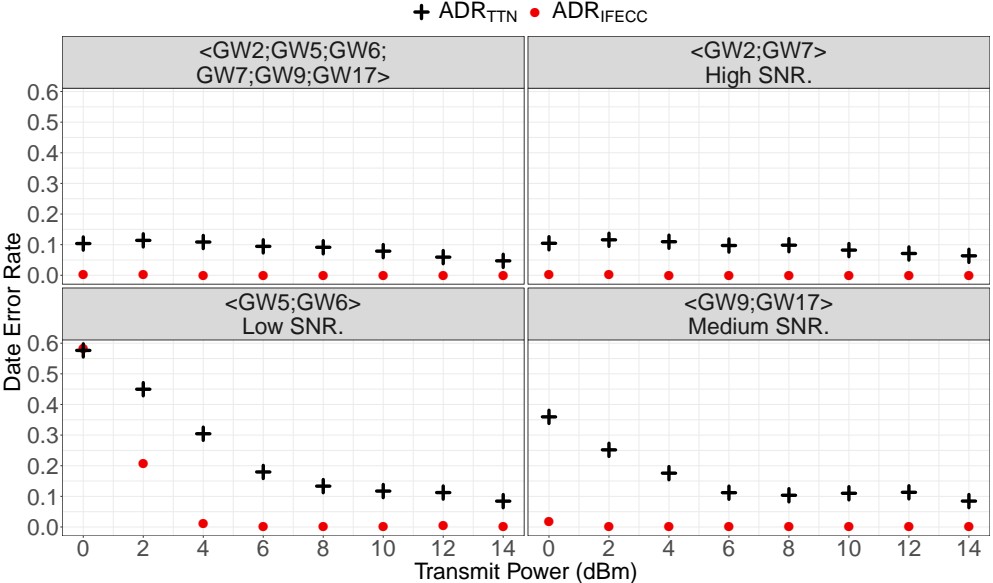

**Figure 19.** *DER* as a function of $P_{Tx}$, for selected real world series of frames replays.

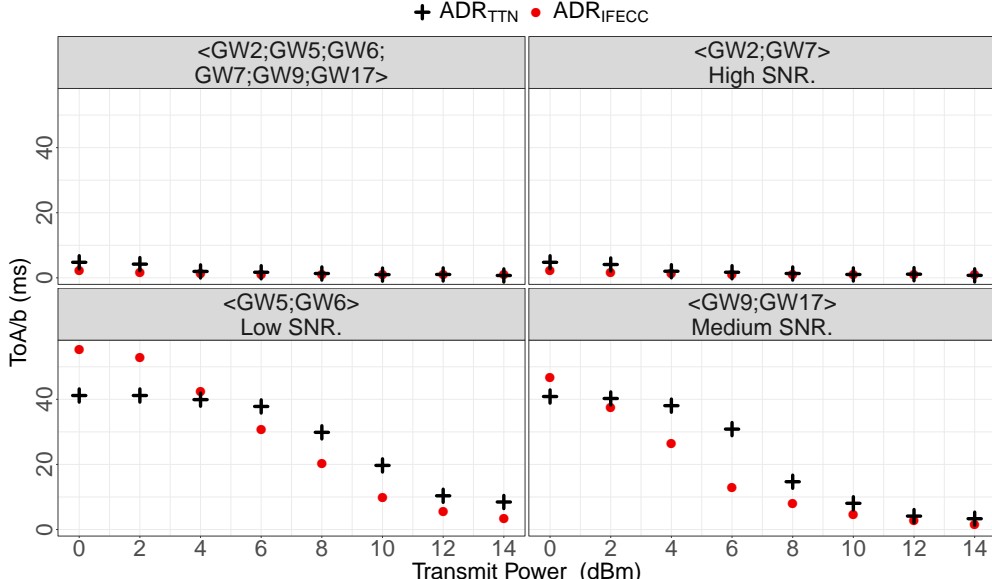

**Figure 20.** *ToA/b* as a function of $P_{\text{Tx}}$, for selected real world series of frames replays.

## 5. Discussion

### 5.1. Adaptive Data Rate

Various studies evaluate and improve the ADR's performances. But because the algorithm is not strictly defined by the LoRaWAN specification, various implementations exist and variations of their interpretation appear in the literature. Some studies [17,18] suggest that the $ADR_{\text{TTN}}$ tends to overestimate the link quality because of the MAX operator used for the *SNR* estimation. As a consequence, they suggest to replace it by a MEAN operator. But because the packets with lowest *SNR* are likely to be more censored, the current path loss' estimation can be biased by both MEAN and MAX. Moreover, the *SNR* variance has a major influence on the ADR's operation [18]. We think that the *SNR* distribution pattern and parameters estimation as described in Section 4 are key for optimized ADR decisions.

The ADR can be slow to converge, especially decreasing to more robust and lower Data Rate, because it relies on EDs to drift toward more robust configuration until a downlink is received [19,20]. For the same reason, the ADR does not converge to the same final configuration depending on its starting configuration. Our solution is no longer based solely on this drift to switch to a more robust configuration.

In a previous work [12], we improved the ADR protocol by relying on the characterization of the channel as a Rayleigh channel and the use of an application layer FEC algorithm. This solution is only tailored for a single cell LoRaWAN network, which is a major weakness for dense deployments composed of few to many gateways. Our new solution is based on a more accurate estimation of the effective channel and dynamically adapts to the number of GWs in range and fully exploits the macro diversity, making it more suitable for real world deployment.

The ADR algorithm can be extended as a load-balancing algorithm to maximize the overall throughput on a single cell LoRaWAN network [21] but this may come at the cost of decreasing the network's reliability. A load-balancing algorithm can balances the traffic intensity in each *SF* virtual sub-channel which reduces the overall collisions in a single cell network [22], and can be extended to $P_{\text{Tx}}$ allocation to reduce near-far problems in the network [23], or extended to the multiple gateways case [24,25]. These studies optimize *SF* allocation through Integer Linear Programming (ILP) considering only the traffic intensity for each SF and they do not take into account capture effect, nor LoRa inter-SF imperfect orthogonality. Moreover, they are based on simplistic channel path loss models that, in particular, do not reflect the magnitude of temporal fading which we believe impacts

significantly capture effect and inter-SF interference. However, optimal static allocation of *SF* may take into account contention with capture effect, a realistic path loss model with temporal fading over various LoRaWAN networks topology such as wide and sparse, or small and dense, mono-gateway cells [26]. An ILP formulation allows to solve the case with inter-SF interference and multi-gateways networks [27]. Notice that these solutions come at the expense of an increased ToA, and thus energy consumption, for some nodes that are set to use higher *SF* than required in order to reduce the collision rate. The ADR algorithm can be set to balance this energy consumption overhead and maximize the network lifetime [28].

An algorithm to select adequate LoRa transmissions parameters to achieve a given reliability between one transmitter and one receiver while reducing energy consumption has been proposed [29]. It starts from the most robust setting and evolves towards a satisfactory setting after the transmission of a few hundreds probes while temporal dynamics is handled by regular restarts. All of this makes it impractically slow compared to our needs.

### 5.2. LoRa/LoRaWAN Link Characterization

Thanks to LoRa and LoRaWAN academic and industrial interest, many experimental measurements are reported in the literature.

As we consider a context with static EDs and GWs, the Large Scale Fading (LSF) due to the distance and propagation medium path loss exponent between the radios is constant. For the same reason, the Shadow Fading (ShF) from obstructions over the main path is also constant. As a consequence, the variations in the receive signal strength are due to Temporal Fading (TF) which corresponds to the gain from multi-path propagation. We neglect the effect of the ambient noise variations, interference, temporal changes of the propagation medium, fast shadowing due to movements around the receiver and transmitter. Thanks to LoRa and LoRaWAN academic and industrial interest, many experimental measurements are reported in the literature.

Three experimental measurements of LoRa link in outdoor environments [15,30,31] provide insight into real world link quality variations. They observed a standard deviation of respectively 8 dB, 7.1 dB and between 6.9 dB and 11.2 dB of the channel gain. Notice that among these studies, only one takes into account the censoring of the frames received with low receive power [31].

Another experimental study of the LoRa link characterization over a public LoRaWAN network in a medium sized city [11] shows that the frame's size has relatively small impact on the reception rate and highlight the impact of an initial successful synchronization probability. The behavior of their experimental channel *SNR* distribution seems to follow a truncated exponential distribution which is expected from a censored Rayleigh channel. The LoRa channel characterization as Rayleigh is also supported by a different study in the same city [12]. LoRa can also be subject to periodic variation of the link quality: an experimental study exposes a periodic 20 dB fading over a 10 km LoRa transmissions that may be caused by daily variation of the air's refraction index combined to multi-path propagation [32].

An experimental study of the LoRa indoor path loss in multi-floor buildings, mainly focused on LSF and ShF, provides some insight into TF: up to 20 dB variation might be encountered because of people's movement [33]. Notice that the TF measurement is fit into a Rician distribution, corresponding to multi-path propagation with a dominant path, but this result is to be taken carefully since it is the signal's envelope, and not the received power, that is expected to follow Rician distribution [34,35]. Similarly, TF measurement is compared to a Rayleigh distribution. Again, the Rayleigh distribution corresponds to the expected signal envelope's distribution in the case of multi-path without dominant path. In this case, the received power is expected to be exponentially distributed [35]. However, no information is given on the packet loss during the TF measurement and a censored data set might results in false positive distribution model fitting as we discuss in Section 2.3.

However, the fact that the people's movement highly increases the LoRa TF is also briefly confirmed by another experimental study [36].

## 6. Conclusions

In this paper, we establish a channel model which reflects the observations we make after generating and analyzing traffic collected on a real-word LoRaWAN deployment. The channel model includes the distribution of the received power around its average, the erasure patterns, and the influence on the demodulation floor of the frame length, as well as of LoRa Forward Error Correction.

We then use this model to derive the expected *PER* for any transmission parameter settings in presence of an arbitrary amount of GWs. This *PER* prediction is the basis of $ADR_{\mathrm{opt}}$, an Adaptive Data Rate mechanism, which can be configured to match an arbitrary reliability goal in terms of Packet Error Rate. $ADR_{\mathrm{opt}}$ inherently takes into account macro-diversity and the observed channel variability due to temporal fading. We also build $ADR_{\mathrm{IFECC}}$ which efficiently provides high reliability, with *Data Error Rate* $< 0.01$ in LoRaWAN networks, even for challenging transmission conditions. It is a significant improvement over the LoRaWAN ADR implemented by *The Things Network*. $ADR_{\mathrm{IFECC}}$ tackles the inevitable erasures of LoRa communications by using an Inter-Frame Erasure Correction Code. $ADR_{\mathrm{IFECC}}$ does not necessitate any additional downlink transmissions compared to legacy LoRaWAN ADR. Both $ADR_{\mathrm{opt}}$ and $ADR_{\mathrm{IFECC}}$ Time on Air are bounded by the maximal effort configuration. They are designed with scalability in mind and are realistic options for current and future deployments. The $ADR_{\mathrm{opt}}$ and $ADR_{\mathrm{IFECC}}$ propositions are validated both by simulation and by replaying actual transmission traces.

**Author Contributions:** Conceptualization, U.C., M.H. and B.T.; Data curation, U.C.; Formal analysis, U.C., M.H. and B.T.; Investigation, U.C. and M.H.; Methodology, U.C., M.H and B.T.; Software, U.C.; Supervision, M.H. and B.T.; Validation, U.C., M.H. and B.T.; Visualization, U.C.; Writing–original draft, U.C.; Writing–review and editing, M.H. and B.T. All authors have read and agreed to the published version of the manuscript.

**Funding:** This work has been partially supported by the French Ministry of Research project PERSYVAL-Lab under contract ANR-11-LABX-0025.

**Institutional Review Board Statement:** Not applicable.

**Informed Consent Statement:** Not applicable.

**Data Availability Statement:** Data available in a publicly accessible repository that does not issue DOIs (Publicly available datasets were analyzed in this study. This data can be found here: https://gricad-gitlab.univ-grenoble-alpes.fr/coutaudu/lora-measurements).

**Conflicts of Interest:** The authors declare no conflict of interest.

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
