# Peer review of "LoRa Channel Characterization for Flexible and High Reliability Adaptive Data Rate in Multiple Gateways Networks"

_computers, doi:10.3390/computers10040044_

Round 1
Reviewer 1 Report
Lorawan protocol is a novel IoT protocol that is expected to provide high reliability and flexible communication links in low energy consumption environments.
The proposed algorithm has good reliability and flexibility. A large number of experimental data and simulation results show that the algorithm is effective and meets the most stringent reliability conditions defined by the industry.
Considering that the LoRaWan protocol must meet the requirements of low energy consumption, and the improved scheme and algorithm should have low energy consumption, the algorithm in this paper lacks more explanation on the impact of energy consumption, so it is suggested to add a certain length to explain.
Reviewer 2 Report
This article suggests a current and attractive topic for the academy. The research is timely and worthwhile. The research problem is clearly defined. The authors provide fresh insight into the field.
I hope you find the following observations helpful:
Structure: Articles should be reformatted according to a standard structure, which is set out in the instructions for authors of the journal (sections are Introduction, Materials and Methods, Results, and Discussions, Conclusion). See new template. The abstract should be improved.
Results: Perhaps it is better to visualize in more charts based on statistical methods of calculation. In my opinion, it may be better to provide the results of testing these methods (if any) in the Results section.
The authors should be appropriate to explain the choice of methodology a little better. It would be advisable to extend the literature review.
Need to revise and check citations in the text and in the references section. I suggest you add this reference: Fedushko S., Ustyianovych T. (2021) Operational Intelligence Software Concepts for Continuous Healthcare Monitoring and Consolidated Data Storage Ecosystem. Advances in Computer Science for Engineering and Education III. ICCSEEA 2020. Advances in Intelligent Systems and Computing, vol 1247. Springer, Cham. pp. 545-557. https://doi.org/10.1007/978-3-030-55506-1_49
Congratulations on a job well done.
Author Response
Please see the attachment.
Thanks for your review and suggestions.
We detailed our response and modification to our article in the attachment.
